# Early–middle Permian Mediterranean gorgonopsian suggests an equatorial origin of therapsids

Rafel Matamales-Andreu [1,2] ✉, Christian F. Kammerer [3], Kenneth D. Angielczyk [4], Tiago R. Simões [5], Eudald Mujal [2,6], Àngel Galobart [2,7] & Josep Fortuny [2]

Therapsids were a dominant component of middle–late Permian terrestrial ecosystems worldwide, eventually giving rise to mammals during the early Mesozoic. However, little is currently known about the time and place of origin of Therapsida. Here we describe a definitive therapsid from the lower–?middle Permian palaeotropics, a partial skeleton of a gorgonopsian from the island of Mallorca, western Mediterranean. This specimen represents, to our knowledge, the oldest gorgonopsian record worldwide, and possibly the oldest known therapsid. Using emerging relaxed clock models, we provide a quantitative timeline for the origin and early diversification of therapsids, indicating a long ghost lineage leading to the evolutionary radiation of all major therapsid clades within less than 10 Myr, in the aftermath of Olson's Extinction. Our findings place this unambiguous early therapsid in an ancient summer wet biome of equatorial Pangaea, thus suggesting that the group originated in tropical rather than temperate regions.

Therapsida is a clade of diverse and ecologically successful tetrapods, with mammals as their modern representatives[1]. The roots of the clade extend back to the late Palaeozoic, when non-mammalian therapsids were important components of terrestrial ecosystems. Until now, the oldest known unequivocal therapsid was *Raranimus dashankouensis*, from probable Roadian (lower middle Permian) deposits of central-east Asia[2]. Yet, phylogenetic analyses consistently suggest that therapsids are the sister group of sphenacodontid 'pelycosaur'-grade synapsids, which originated in the Pennsylvanian (ca. 320 Ma)[3,4]. This implies a long therapsid ghost lineage spanning about 40 million years.

This vast knowledge gap likely stems from the uneven geographic sampling of Permian tetrapods worldwide. The best-known early Permian tetrapod-bearing formations in North America (southwestern 'Red Beds') and in Europe (Central European Basin) correspond to palaeotropical ecosystems and have not yet yielded any definitive therapsid fossils[1,5]. By contrast, the sites with the most diverse and abundant middle to upper Permian therapsid records are in the Cis-Urals of Russia and in southern Africa, preserving ecosystems of northern and southern palaeotemperate latitudes, respectively[6,7]. The historical lack of a spatiotemporal link between these regions has obscured the early history of therapsid evolution and diversification. Furthermore, the exact timing for the origin of the major groups of therapsids has always relied on a literal reading of the fossil record[8], without implementation of statistical modelling.

Here we describe the earliest unequivocal therapsid from equatorial Pangaea, collected in the upper Cisuralian (lower Permian)–? lowermost Guadalupian (middle Permian) of Mallorca (Balearic Islands, western Mediterranean)[9] (Figs. 1, 2). We discuss its identity, age

[1]MUCBO | Museu Balear de Ciències Naturals, FJBS-MBCN, ctra. Palma-Port de Sóller km 30.5, 07100 Sóller, Mallorca, Illes Balears, Spain. [2]Institut Català de Paleontologia Miquel Crusafont (ICP-CERCA), Universitat Autònoma de Barcelona, Edifici ICTA-ICP, c/ Columnes s/n, Campus de la UAB, 08193 Cerdanyola del Vallès, Barcelona, Catalunya, Spain. [3]North Carolina Museum of Natural Sciences, 11 W. Jones Street, Raleigh, NC 27604, USA. [4]Field Museum of Natural History, 1400 South Lake Shore Drive, Chicago, IL 60605, USA. [5]Department of Ecology and Evolutionary Biology, Princeton University, Princeton, NJ 08540, USA. [6]Staatliches Museum für Naturkunde Stuttgart, Rosenstein 1, 70191 Stuttgart, Germany. [7]Museu de la Conca Dellà, c/ del Museu 4, 25650 Isona i Conca Dellà Lleida, Spain. ✉e-mail: rmatamales@mucbo.org

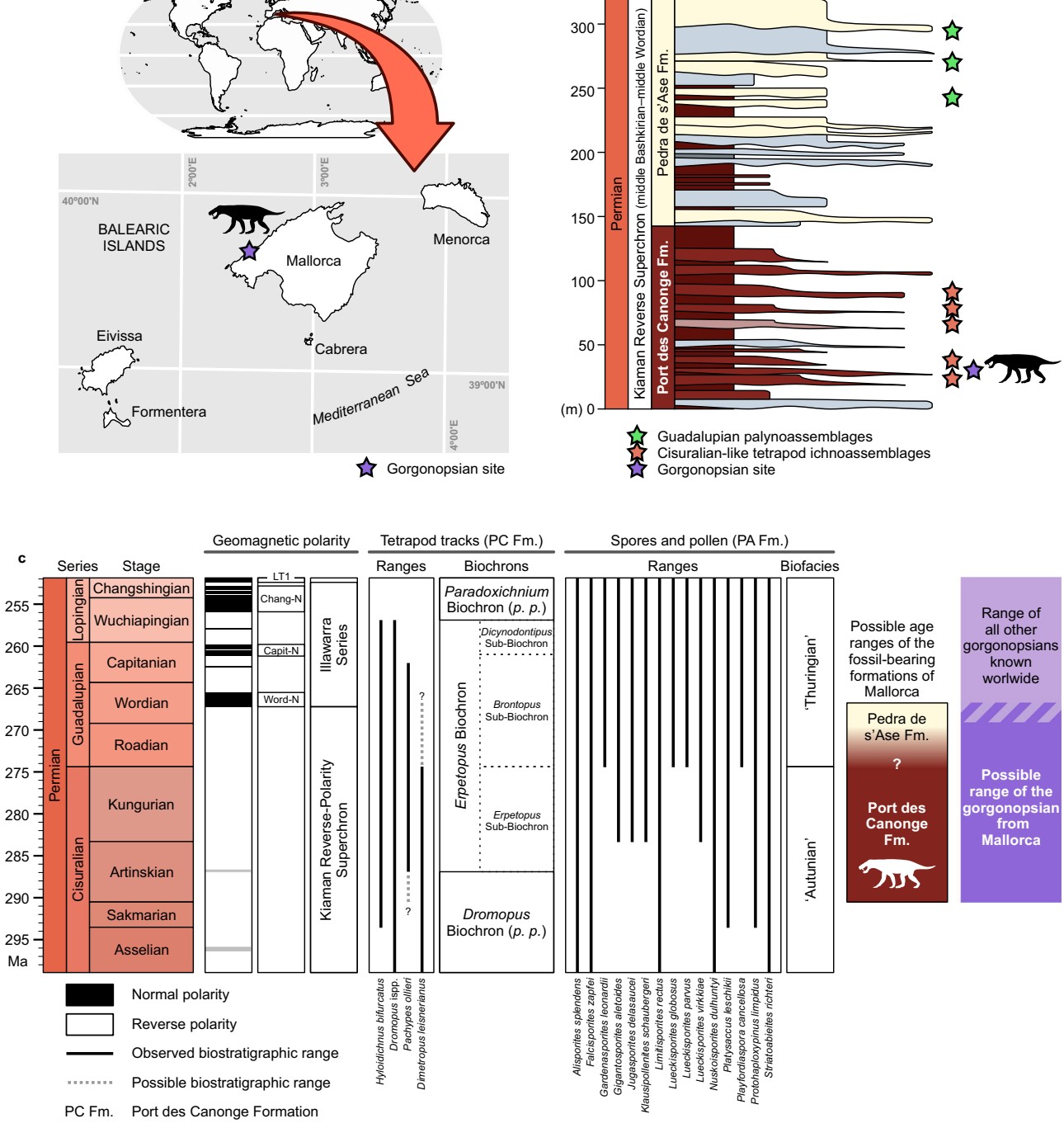

**Fig. 1 | Geological context of the gorgonopsian from Mallorca. a** World map showing the location of the island of Mallorca, and map of the Balearic Islands with indication of the study area. **b** Composite stratigraphic log for the Racó de s'Algar-Pedra de s'Ase section (simplified from Matamales-Andreu et al.[9]), indicating the beds that contain tetrapod tracks, spores and pollen, and the gorgonopsian. **c**, Magneto- and biostratigraphic criteria used to date the Permian formations of Mallorca (modified from Matamales-Andreu et al.[9], with additional data from Marchetti et al.[36]).

and palaeobiogeographical implications, revealing that it is the oldest known gorgonopsian to our knowledge (Fig. 1c), and very likely the oldest therapsid worldwide. The presence of a derived therapsid at this age and location is unexpected, indicates that the diversification of the major therapsid subclades may not have been restricted to high palaeolatitudes as previously thought, and provides insight on the timing and phylogenetic structure of the therapsid evolutionary radiation. Olson's Extinction, which happened in the early–middle

Permian transition[10–12], may have provided the necessary ecological opportunity for this previously inconspicuous group to diversify into varied phenotypes and distinct ecological roles.

## Results and discussion
### Systematic palaeontology
Synapsida Osborn, 1903[13]
    Therapsida Broom, 1905[14]

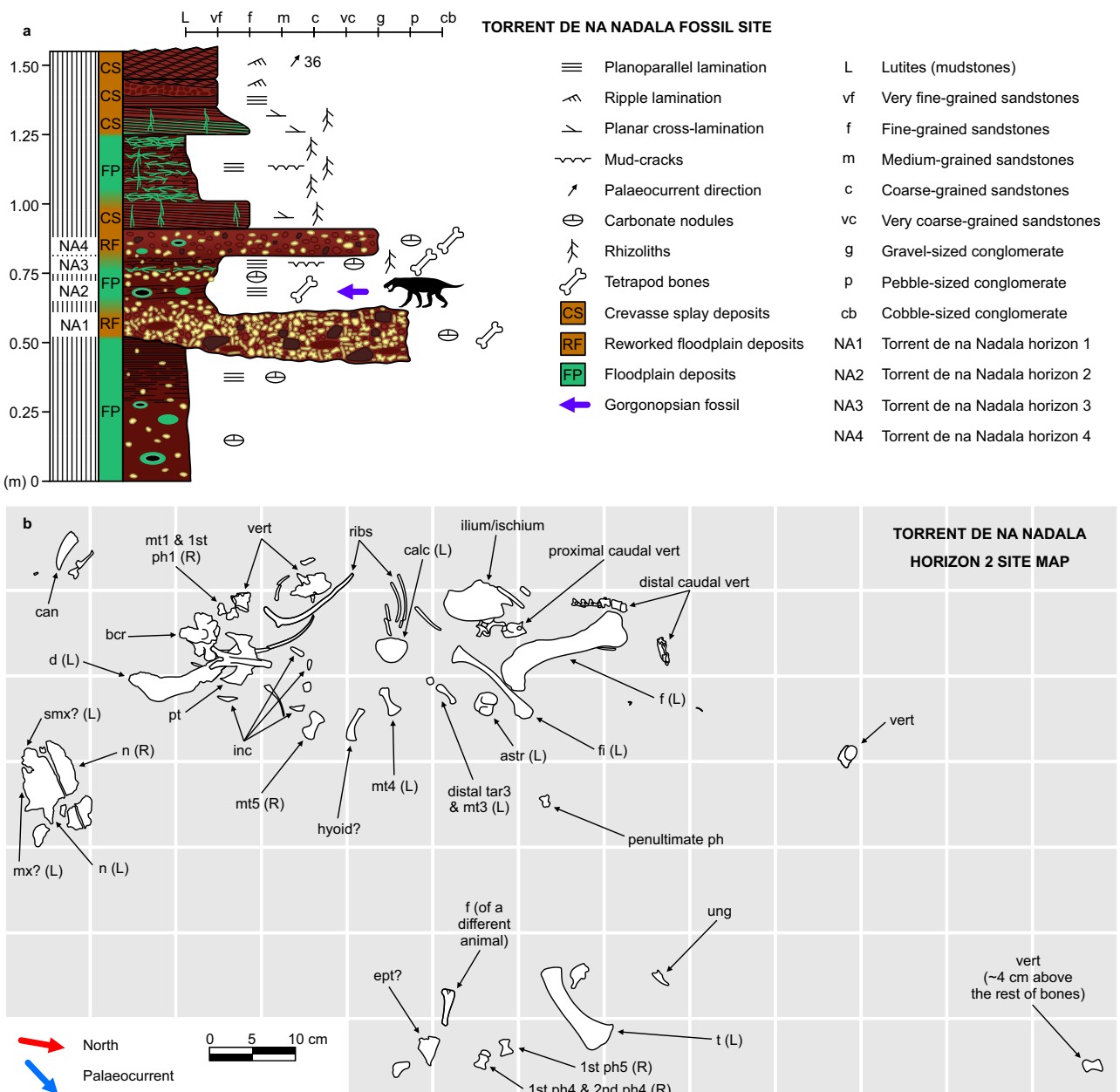

**Fig. 2 | Stratigraphy and site map. a** Detailed stratigraphic log of the Torrent de na Nadala fossil site with indication of the lithology, palaeoenvironment and location of fossiliferous horizons[16]. **b** Site map of the Torrent de na Nadala horizon 2, with all the bones attributed to the gorgonopsian. astr = astragalus, bcr = basicranium, calc = calcaneum, can = canine, d = dentary, ept = ectopterygoid, f = femur, fi = fibula, inc = incisor, (L) = left element, mt = metatarsal, mx = maxilla, n = nasal, ph = phalanx, pt = pterygoid, (R) = right element, smx = septomaxilla, t = tibia, tar = tarsal, ung = ungual, vert = vertebra.

---

Gorgonopsia Seeley, 1894[15]
Gorgonopsia indet.
(Figure 3, Supplementary Figs. 1–5)

**Studied material.** DA21/17-01-01 (see the full list of institution acronyms in Supplementary Note 1), a disarticulated, partial skeleton consisting of partial snout roof, partial basicranium, pterygoid, ectopterygoid?, hyoid?, left dentary, one upper canine, four lower? incisors, four presacral vertebrae, two proximal caudal vertebrae, eight distal caudal vertebrae, two right dorsal ribs and fragments of others, left femur, left tibia, left fibula, left calcaneum, left astragalus, possible left distal tarsal 3, left metatarsals 3 and 4, right metatarsals 1 and 5, first phalanges of right digits I, IV and V, second phalanx of right digit IV, one penultimate phalanx of digit III, IV or V, and one ungual phalanx. All the bones were found near the base of bed NA2 (Torrent de na Nadala 2, Banyalbufar, Mallorca, western Mediterranean[16]; see Fig. 2) except for one of the presacral vertebrae, which was extracted from the upper part of the same bed (ca. 4 cm above), in a conglomerate horizon (Fig. 2b).

### Description and comparative anatomy
DA21/17-01-01 represents a small gorgonopsian (estimated skull length ~18 cm) known from various cranial fragments, several sections of the axial column (including a large portion of the tail), and most of the left hind limb. See Supplementary Figs. 1–5 and Supplementary Note 2 for a detailed description and illustration of all the elements.

Cranial elements consist of a highly-flattened dorsal portion of the snout, fused basicranium incorporating at least the basioccipital,

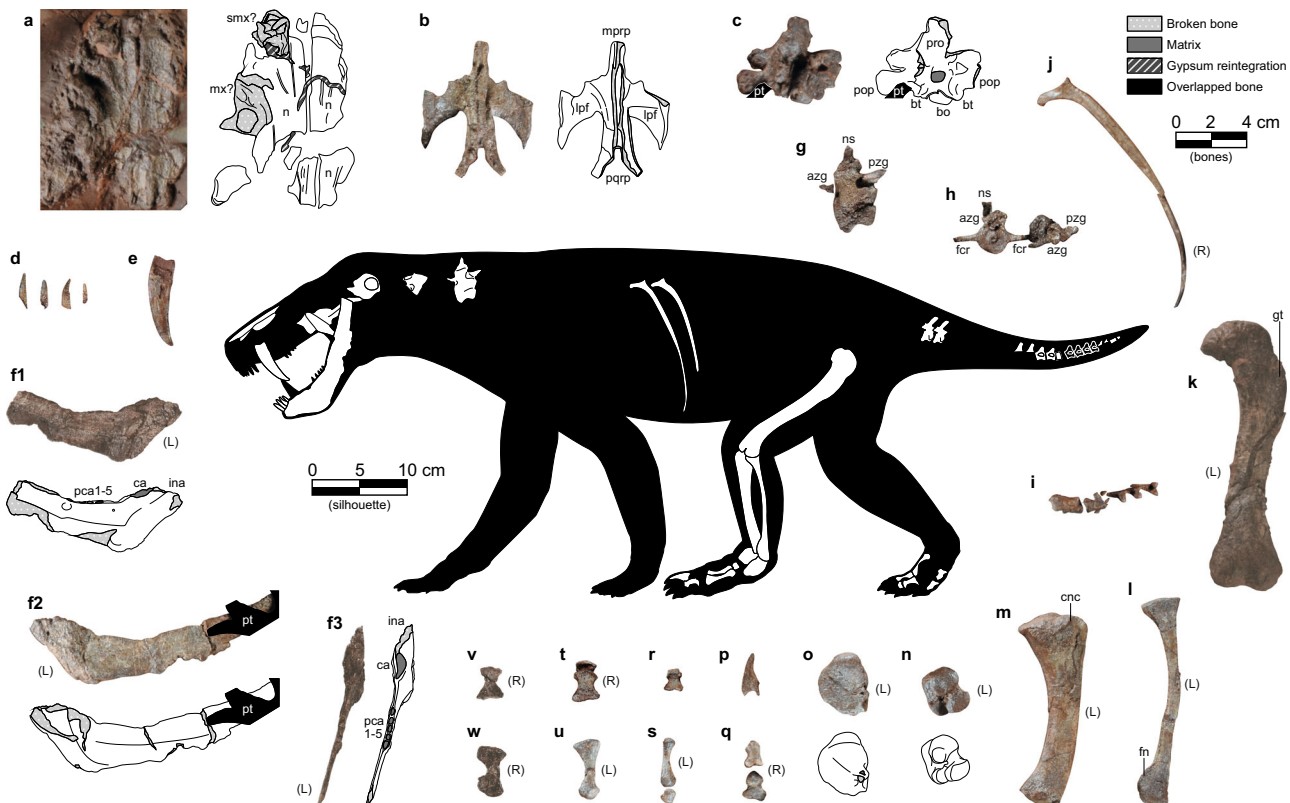

**Fig. 3 | Most relevant elements of DA21/17-01-01 and silhouette showing their positions (see also Supplementary Figs. 1–5). a** Snout fragment. **b** Pterygoid. **c** Basicranium. **d** Incisors. **e** Canine. **f1**-3 Left dentary. **g** Cervical? vertebra. **h** Proximal caudal vertebrae. **i** Distal caudal vertebrae. **j** Right dorsal rib. **k** Left femur. **l** Left fibula. **m** Left tibia. **n** Left astragalus. **o** Left calcaneum. **p** Ungual. **q** First metatarsal (below) and first phalanx of right digit I (above). **r** Penultimate phalanx. **s** Distal tarsal (below) and metatarsal (above) of left digit III. **t** First (below) and second (above) phalanges of right digit IV. **u** Left fourth tarsal. **v** First phalanx of right digit V. **w** Right fifth metatarsal. azg = prezygapophysis, bo = basioccipital, bt = basal tuber, ca = alveolus of canine, cnc = cnemial crest, fcr = fused caudal ribs, ina = alveoli of incisors, lpf = lateral pterygoid flange, mprp = median palatal ramus of the pterygoid, mx = maxilla, n = nasal, ns = neural spine, pca = alveoli of post-canines, pop = paroccipital process (opisthotic), pqrp = pterygoid posterior quadrate rami, pro = prootic, pt = pterygoid, pzg = postzygapophysis, smx = sep-tomaxilla, tp = transverse process, fn = fibular notch, gt = greater trochanter. (L) and (R) next to the elements refer to the left and right sides of the skeleton, respectively.

opisthotic, and prootic, and an isolated pterygoid. The fused pterygoid exhibits prominently 'wing'-shaped, backswept transverse processes (seen in other early gorgonopsians, such as *Eriphostoma* and *Gorgonops*[17]) and bears anterior articular facets for the ectopterygoids. The left dentary is preserved almost in its entirety, with alveoli indicating a tooth count of i4/c1/pc5 (incisor/canine/postcanine). The dentary has a prominent mentum at the ventral edge of the symphysis and a clear inflection point dorsally where the incisor roots would have been housed. Several serrated, weakly spatulate teeth probably represent the lower incisors. A much larger, laterally compressed, blade-like tooth represents a canine, probably an upper based on size.

Preserved axial elements include two probable cervical vertebrae, two intact dorsal ribs and fragments of additional ribs, two isolated anterior caudal vertebrae, and an articulated caudal series including evidence for 14 vertebrae. The vertebrae are similar to those known in other gorgonopsians[18]. The left hind limb includes the femur, tibia, fibula, astragalus, calcaneum, and several metatarsals and phalanges, including one ungual. The tibia and fibula are both bowed, indicating the presence of a large interosseous space. The femur is weakly curved with a notably offset head, which is the typical morphology of small gorgonopsians[19]. In general, the postcranial anatomy of the gorgonopsian from Mallorca is very consistent with that of other small-bodied members of the group (*e.g.*, *Cyonosaurus*), with no clear autapomorphies.

A suite of characters clearly demonstrates that specimen DA21/17-01-01 represents a gorgonopsian therapsid. The dentary is characteristic for this clade, showing the combination of a steeply inclined mandibular symphysis, canine-postcanine diastema, relatively short postcanine tooth row, and anterodorsally-to-posteroventrally angled postcanine tooth row. Together, this combination of characters is only known in Gorgonopsia among Permian synapsids[19]. A steep mandibular symphysis is also present in anteosaurs[20], and some biarmosuchians and therocephalians (e.g., *Hipposaurus*[19], *Moschorhinus*[21]), but in these taxa no diastema exists between the lower canine and the (usually extensive) postcanine tooth row. In most therocephalians and some biarmosuchians (e.g., *Herpetoskylax*[22]), the symphysis is generally weakly sloping, with a low, convex anterior face[23]. Anterodorsal-to-posteroventral angulation of the postcanine tooth row is typical of biarmosuchians and therocephalians in addition to gorgonopsians, but this character excludes identification as a dinocephalian or a non-therapsid synapsid ('pelycosaur'). Furthermore, the triangular anterodorsal projection housing the incisor tooth row is seen in almost all small gorgonopsians (Sigogneau[24]: fig. 206). The canine is mesiodistally serrated and very compressed labiolingually (although this may be exaggerated by taphonomic distortion). This tooth is more circular in cross-section in other early synapsid groups with serrated canines (biarmosuchians, anteosaurs, and early therocephalians; also the caniniforms of some sphenacodontids)[25]. The pterygoid shows a backswept 'wing'-like morphology of the transverse processes, which is frequently observed in early gorgonopsians and differs strongly from the more robust-edged transverse processes of biarmosuchians and dinocephalians, which lack attenuate tips. Some early therocephalians also have backswept transverse processes of the

pterygoid[23], but in therocephalians the ectopterygoid is located lateral to this process, such that there is not an anterior facet for the ectopterygoid as in DA21/17-01-01. The general morphology of the femur (sinuous with the head sharply offset from the shaft; low, rounded fourth trochanter; and greatly expanded distal end) is characteristically gorgonopsian, and the fibula and especially the tibia bow strongly away from one another (indicating a large interosseous space), a morphology unique to gorgonopsians among therapsids[26].

Because of the incompleteness of the specimen, it is difficult to identify it more precisely than Gorgonopsia, especially considering the notorious morphological conservatism of the clade. Some types of gorgonopsians can clearly be excluded as possibilities, however. The morphology of the pterygoids is unlike that of Rubidgeinae and related taxa (*e.g.*, *Arctops*), in which the transverse processes are generally swollen towards their tips and not backswept[27,28]. The presence of five lower postcanines also excludes several rubidgeine taxa (*e.g.*, *Clelandina*, *Rubidgea*) as well as various other gorgonopsians with absent (*e.g.*, *Inostrancevia*) or reduced (*e.g.*, *Eriphostoma*) lower postcanine tooth rows[17]. However, the same tooth count is known in small gorgonopsians such as *Aelurosaurus felinus*[24]. A variety of other features distinguish this specimen from rubidgeines and *Inostrancevia*, but these may be more size-related than strictly phylogenetic. For example, the inflected anterior face of the mandibular symphysis and the sinuous femoral shape differ from the straight, massive symphysis and nearly-straight femur of large gorgonopsians, but they are known in most small-bodied members of the clade regardless of relationships[24]. Ontogenetic variation in these features is poorly understood, and whether they change with increasing body size during the life of an individual is currently unknown. However, this specimen likely represents an osteologically mature individual, based on the very well-ossified limb and ankle elements and fused basicranium (Fig. 3, Supplementary Fig. 1), instead of a juvenile of one of the larger species (those with skull length 40 cm or greater). Although the exact size of the animal is hard to determine given the paucity of elements and limited basis for postcranial comparisons in the group, based on dentary length, and assuming comparable proportions to *Gorgonops*, this individual would have had a skull length of ~18 cm (Fig. 3).

Presence of a steep mandibular symphysis suggests that the specimen from Mallorca is more derived than the current earliest-diverging gorgonopsian, *Nochnitsa* (following the character polarity of Kammerer & Masyutin[29]), but otherwise its cranial characters are generally plesiomorphic. The elongate, backswept transverse processes of the pterygoids are seen in *Eriphostoma*, *Gorgonops* and *Cynariops*[17,30] among African taxa, but also in the earlier-diverging *Viatkogorgon*[29], suggesting that this morphology is ancestral for Gorgonopsia. The relatively anterior position of the divergence between the quadrate rami of the pterygoids is also observed in *Gorgonops*, *Aelurosaurus*, and *Cyonosaurus*, and the close proximity between the basal tubera and the occipital condyle is comparable to that of *Gorgonops*, *Suchogorgon*, and *Sauroctonus* (Kammerer, pers. obs.). Overall, the gorgonopsian from Mallorca shows features most consistent with early-diverging members of the 'African clade' (*sensu* Kammerer & Masyutin[29]), but, as these are plesiomorphic, it could also fall outside of this subclade as either an early-diverging member of the 'Russian clade' (like *Suchogorgon*) or outside of the major divergence altogether (like *Viatkogorgon*). Given its stratigraphic and geographic separation from other members of the group, this specimen likely represents a new taxon, but in the absence of any uniquely diagnostic features in the preserved material, we do not erect a new species for it here.

## Age of the oldest known therapsids
The bone-bearing beds containing the gorgonopsian described here have been argued to belong to the Artinskian or Kungurian stages of the Cisuralian series (lower Permian)[9], based on four lines of

evidence. First, magnetostratigraphic analysis showed that the uppermost levels of the succession, 250 m stratigraphically above the bone-bearing levels, belong to the Kiaman Reverse Superchron, which dates from the middle Bashkirian to middle Wordian (early Pennsylvanian to middle Permian)[31]. This indicates that the gorgonopsian from Mallorca must be older than middle Wordian. A pre-Wordian age for the specimen is very likely considering the thick overlying sequence of mudstones with carbonate palaeosols (Fig. 1b)[9], a type of rock with low sedimentation rates[32]. Second, palynoassemblages from the upper part of the overlying Pedra de s'Ase Formation provided additional evidence for a Guadalupian (middle Permian) age for the uppermost part of the section (Fig. 1b, c). Third, the tetrapod tracks stratigraphically bracketing the gorgonopsian site (Fig. 1b, c) possess the typical components of Cisuralian ichnoassemblages elsewhere (Fig. 1c), including the ichnogenus *Dimetropus*, which is unknown from rocks younger than the Kungurian (upper part of the lower Permian)[33]. 'Pelycosaur' body fossils are known from the Kungurian to the Roadian of Russia[34], North America[5], and Sardinia[35], raising the possibility that this ichnogenus could be found in Guadalupian rocks, but the abundance of 'pelycosaur'-dominated ichnoassemblages in the Port des Canonge Formation is more typical of the Cisuralian worldwide, regardless of the sedimentary palaeoenvironment[33]. Fourth, these ichnoassemblages lack the ichnogenera *Erpetopus* (middle Artinskian–middle Wuchiapingian) and *Brontopus* (Roadian–middle Capitanian), which commonly appear in younger strata in palaeogeographically proximate basins[33,36,37].

Based on these arguments, the Mallorcan gorgonopsian is certainly older than middle Wordian (early Guadalupian, middle Permian), and it may be as old as Artinskian (late Cisuralian, early Permian) (Fig. 1c). The minimum age places it among the oldest known gorgonopsians worldwide, rivalled only by an indeterminate gorgonopsian from the Wordian *Eodicynodon* Assemblage Zone of South Africa[38]. Because the stratigraphic position of the specimen from Mallorca is 250 m below the last bed of the section, which is still part of the Kiaman Reverse Superchron (Fig. 1c), it is very likely to be older than Wordian, therefore making it the oldest record of gorgonopsians worldwide, to our knowledge. A Roadian age for the Mallorca specimen would make it a contemporary of the oldest therapsids (e.g., *Raranimus*, *Biseridens*) currently known from the Dashankou assemblage of China[2], whereas a Kungurian or Artinskian age, which best accords with the magnetostratigraphic and ichnological data, would make it the oldest definitive therapsid discovered to date.

## Phylogeny and a detailed timeline for the origin and early radiation of therapsids
Currently, no definitive early Permian therapsids are known, despite widespread agreement that the divergence between therapsids and other synapsids must have happened by that time (and possibly as early as the Pennsylvanian[4,39]). The unusual Artinskian synapsid *Tetraceratops* has been proposed to partially fill this gap[40,41], but more recent research suggests that it is a non-therapsid eupelycosaur[42], which is supported by our phylogenetic results (Fig. 4, Supplementary Fig. 6). This important fossil gap in the early history of therapsids has historically obscured our understanding regarding the time of origin of Therapsida and its major clades. Furthermore, discussions of therapsid origins have mostly been conducted by qualitative assessments and literal readings of the therapsid fossil record, which have been used to argue for an explosive radiation of the clade in the middle Permian[8].

To advance our knowledge on this topic, we used, for the first time, mechanistic models of evolution to estimate divergence times for the origin of the major therapsid clades; namely Bayesian relaxed morphological clock analyses. We used the skyline fossilised birth-death tree model (SFBD)[43] in combination with new relaxed clock

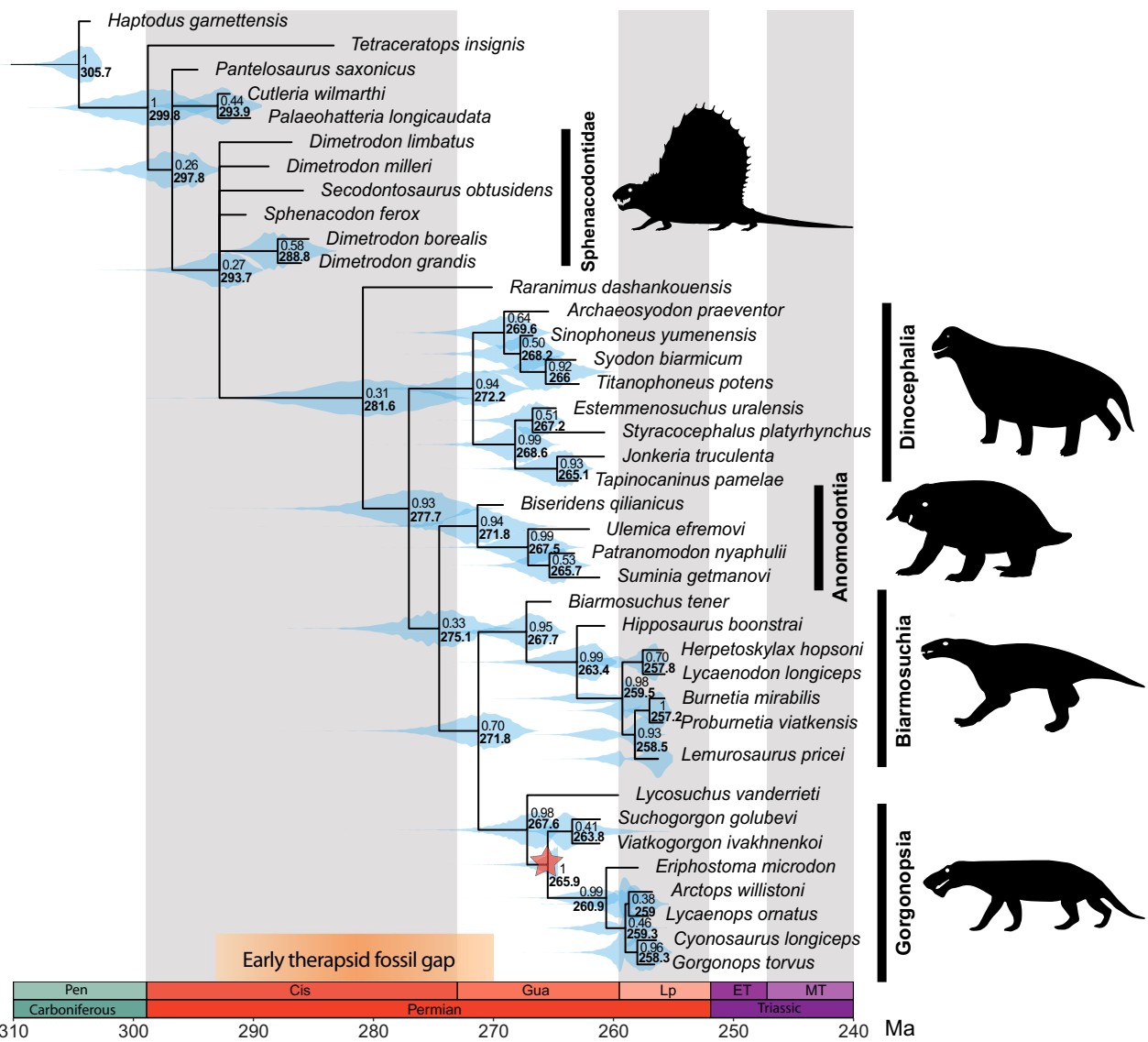

**Fig. 4 | New time-calibrated evolutionary tree for the major groups of early therapsids.** Maximum compatible tree from relaxed morphological clock Bayesian inference analysis under the skyline fossilised birth-death tree model (SFBD) using tip-dating and the best fitting clock model (see Methods and Supplementary Table 1). All the major therapsid groups originated much later than 'total group' therapsids, but radiated in a relatively short time span of only ca. 10 Myr. Red star indicates single node calibration based on new Mallorca specimen (see Supplementary Fig. 6 for variance in divergence time estimates with and without new data from the Mallorcan site). Node values represent posterior probabilities (top) and median ages (bottom, in bold). Node violin plots (cyan) represent the distribution of the 95% highest posterior density (HPD) intervals. All silhouettes produced by TRS.

models recently implemented in the software Mr. Bayes[44] – see Methods. We updated and expanded the phylogenetic data matrix of Brink et al.[45] to create the most comprehensive available dataset sampling the major early therapsid clades at the species level. We implemented tip-dated calibrations followed by a tip + node-dating. In the latter, we applied a node calibration for the age of Gorgonopsia informed by the range of potential ages of the Mallorcan specimen to see its impact on divergence time estimates.

The results of this analysis provide an updated temporal framework for the therapsid diversification (Fig. 4, Supplementary Fig. 6). We infer the split between therapsids and 'pelycosaurs' (represented here by non-therapsid sphenacodonts) to have occurred at ~294 Ma (95% HPD = 290.5–298.0 Ma) in the earliest Permian (Asselian), rather than in the late Carboniferous (Pennsylvanian), as previously hypothesised[39]. We confirm the traditional understanding that there was a relatively long ghost lineage of about 15 Myr between the origin of 'total-group' therapsids and the radiation of the major therapsid

clades (dinocephalians, anomodonts, biarmosuchians, gorgonopsians, therocephalians) at ~278 Ma in the late Cisuralian. Our results support *Raranimus* as the only known member of this early ('stem') stock of therapsids before the origin of the major therapsid clades. Despite its incompleteness and relatively young age (early Guadalupian; coeval with dinocephalians and anomodonts), *Raranimus* preserves several morphological characters that would support this phylogenetic interpretation, including two upper caniniform tooth positions and an elongate caudal alveolar portion of the maxillary canal[2].

Interestingly, despite the long ghost lineage in the early history of therapsids, our results support previous qualitative interpretations of the fossil record indicating that major therapsid clades originated over a relatively short span of time, which we infer here to be only 10 Myr, between ~278–268 Ma (late Cisuralian–early Guadalupian, or early–middle Permian transition). This time range encompasses the period of heightened tetrapod turnover known as Olson's

Extinction at ca. 273 Ma[10-12], which marked the extinction of important components of terrestrial tetrapod faunas at the time, including other synapsids (Ophiacodontidae, Edaphosauridae) and early tetrapods (Molgophidae, Trimerorhachidae), besides major drops in diversity for Sphenacodontidae and Eryopidae (i.e., 'dead clades walking'). After this event, the emergence of robust herbivorous lineages, such as pareiasaurs, appears to have prompted a dramatic shift in jaw morphofunctionality of carnivorous synapsids (mostly therapsids) towards injuring and subduing the prey, instead of the optimisation for grasping and holding smaller prey seen in 'pelycosaurs'[46]. The loss of potential competitors and the evolution of novel morphotypes in terrestrial animals may have provided ecological opportunity for emerging therapsid clades, catalysing their evolutionary radiation in the middle Permian.

The Bayesian clock results also inform our expectations for the structure of the early therapsid fossil record. Based on divergence times of the major therapsid clades, we predict that morphologically diagnosable members of these clades could be found in rocks as old as the Kungurian (273–283 Ma). As a diagnostic member of Gorgonopsia whose potential age falls in this range, the Mallorcan specimen corroborates this prediction (Fig. 4, Supplementary Fig. 6). Our revised understanding of the therapsid fossil record also makes it unlikely that we will find diagnostic members of major therapsid clades in rocks older than the Artinskian (~285 Ma). Instead, we predict that therapsids from this time interval would be 'stem' members of the group, lacking many diagnostic characters found in the later clades, and likely difficult to differentiate from contemporaneous 'pelycosaurs'. Critical reappraisal of previously collected fossils (e.g., Spindler[42]), and collecting efforts in under-sampled regions from the continental northwestern Tethyan domain, are important areas of ongoing work in the search for the currently missing 'stem' therapsids.

## Biogeographic origin of therapsids

A long-established pattern in the biogeography of early synapsids is that early Permian 'pelycosaurs' occur in lower palaeolatitudes and middle–late Permian therapsids in higher palaeolatitudes. This geographically disjunct distribution has obscured the biogeographic centre of origin of therapsids[1]. Kemp[8], for instance, hypothesised that the first therapsids would have evolved from sphenacodont 'pelycosaurs' during the late part of the early Permian, acquiring a unique combination of traits that facilitated their occurrence in the seasonally arid tropical biomes of Pangaea. There, they would have persisted as medium-sized carnivores until the middle Permian, when the tropical and temperate biomes became connected, allowing therapsids to diversify and colonise higher latitudes. However, until now, tropical therapsids were rare and only known with certainty from the late Permian (Lopingian), represented by a poor fossil record consisting of fragmentary gorgonopsian remains from central Africa[47] and derived dicynodonts from Laos[48]. Unfortunately, those records provide no relevant information about the time or place of origin of therapsids. Richer therapsid assemblages are known from northern China, but the complex geological history and conflicting palaeogeographic reconstructions for this region complicates confident palaeolatitudinal assignments. Although current evidence indicates that the North China Block was within the tropics during the Permian[49], the portion of Gansu yielding middle Permian therapsids (the possibly Roadian Dashankou assemblage) was part of a distinct land mass (the Alashan Terrane) that did not amalgamate with the North China Block until the Mesozoic and was situated north of it (though at least in the subtropics)[50].

The gorgonopsian from Mallorca provides the first unequivocal evidence that therapsids were indeed present in the summer wet biomes of equatorial Pangaea during the early–middle Permian transition, suggesting that the group may have originated in lower, tropical latitudes, rather than in the higher latitudes where nearly all of their fossils are known (Fig. 5). The Mallorcan specimen represents a specialised form that, based on the fossil track record of its locality —dominated by moradisaurine captorhinids and small-sized 'pelycosaurs'[9,16], was probably the apex predator of its ecosystem. However, contra Kemp's hypothesis[8] that the diversification of therapsids into their major subclades was driven by expansion into seasonally cool temperate regions, the presence of a definitive gorgonopsian (a clade deeply nested in therapsid phylogeny; Fig. 4) in equatorial Pangaea implies that this radiation was underway prior to synapsid dispersal to higher latitudes.

If therapsids originated in the tropics, this has implications for metabolic evolution in the clade. Although recent evidence indicates that mammal-like endothermy in synapsids did not originate until the Triassic[51], the expansion of early therapsids in higher latitudes[8] suggests that, following Olson's Extinction, the members of this clade were more eurythermal due to some rudimentary form of thermoregulation. The palaeotropical gorgonopsian presented here implies that this early thermal niche relaxation was probably the result of an exaptive process, in which early therapsids first developed their revolutionary new suite of morphological and physiological traits (more efficient locomotion, respiration, food manipulation, and elevated growth rates, among others) in the ancestral synapsid equatorial belt, which later facilitated their colonisation of cooler climates. Given that the earliest therapsids were all faunivorous[52], an additional selective force operating on the therapsid body plan may have been the appearance of larger prey that required novel methods to subdue[46]. Large moradisaurine captorhinids became abundant in the Kungurian–Roadian[53,54], with one of these species and large tracks correlated to this clade found in the same formation as the Mallorcan gorgonopsian[9,55], and geographically proximate beds of similar age in Menorca, Sardinia and southern France have yielded enormous moradisaurine captorhinids[56] and caseid 'pelycosaurs'[57]. Possibly, the traits selected to prey upon these animals also benefitted therapsids as they expanded into new environments inhabited by other large tetrapods.

This discovery underscores the need for more sampling in poorly studied areas of northern Africa and southern Europe that preserve Permian tropical ecosystems. Available data suggest that, at least during the late Permian, tropical tetrapod assemblages consisted of an 'atypical' mixture of classical middle–late Permian taxa (pareiasaurs and therapsids) and relict groups otherwise widespread during the late Carboniferous and early Permian (cochleosaurid temnospondyls, diplocaulid lepospondyls, moradisaurine captorhinids)[6,7,58,59]. The assemblage from Mallorca is also unique but in a different sense, as in this case it shows a mixture of typically early Permian tetrapods ('pelycosaurs' and moradisaurine captorhinids) and therapsids. These observations hint at a complex tetrapod palaeobiogeographical history in the tropical regions of Pangaea, with therapsids constituting a key component.

Our findings demonstrate that therapsids inhabited equatorial Pangaea during the early–middle Permian. Specifically, we provide, to our knowledge, the first record of Gorgonopsia in western Europe, represented by a relatively small taxon (Figs. 3, 6). Several independent lines of evidence (palaeomagnetism, palynoassemblages, tetrapod ichnoassemblages) point to a late Cisuralian/?earliest Guadalupian age. This implies that the specimen is the oldest (albeit not earliest-diverging) gorgonopsian worldwide, and is among the oldest therapsids, if not the oldest member of the clade discovered to date. Our Bayesian evolutionary analyses indicate that diversification of the major therapsid clades was underway by the Kungurian, potentially in association with Olson's Extinction. After a long ghost lineage in the early history of Therapsida, its major clades originated in a relatively short span of time of less than 10 Myr between the late Cisuralian and the early Guadalupian (280–270 Ma). Prior to that time, we are likely to only find stem therapsids that had undergone

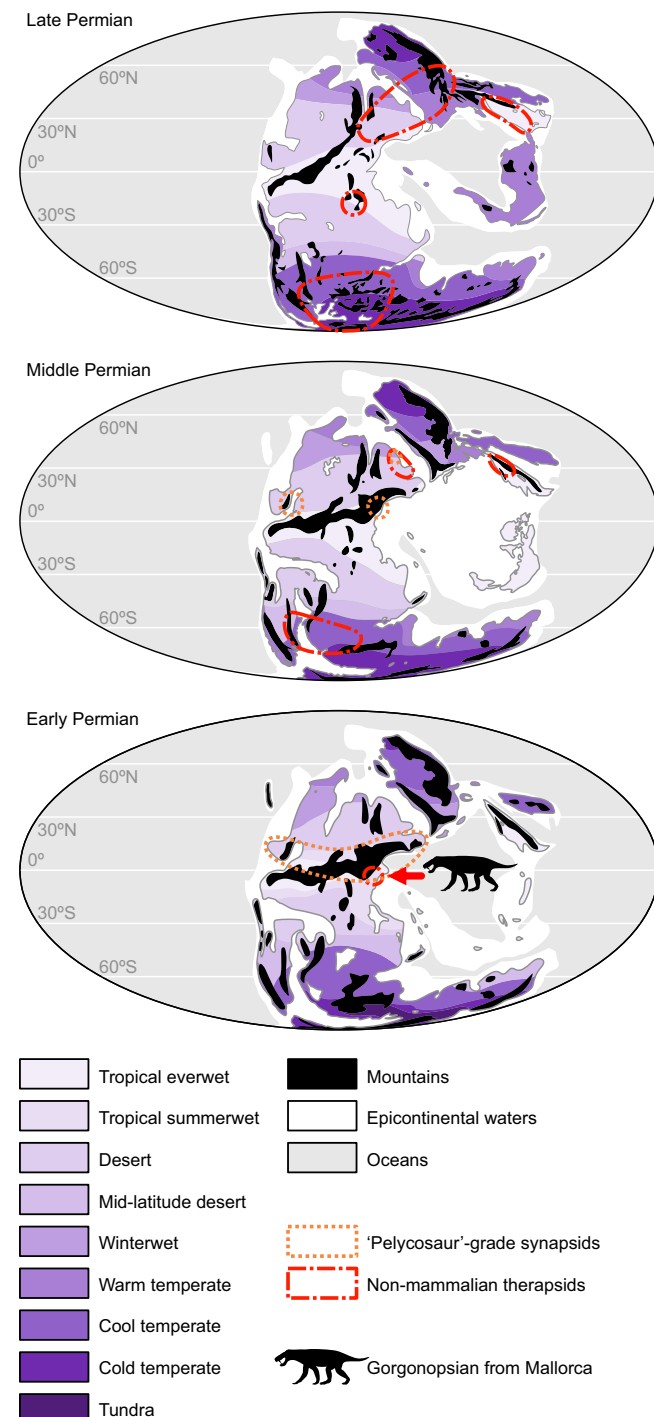

**Fig. 5 | Palaeobiogeographical distribution of Permian synapsids (excluding varanopids) through time.** Note a dominant tropical distribution of 'pelycosaur'-grade synapsids during the early and early part of the middle Permian, and a trans-Pangaean distribution of non-mammalian therapsids during the middle and late Permian. The specimen studied herein has been displayed on the early Permian map because that is its most probable age, although it could also be from the earliest middle Permian. Maps after Scotese[72], and climate zones after Rees et al.[73] for the early and middle Permian, and Smith et al.[60] for the late Permian. Synapsid distribution after Laurin et al.[5], Bernardi et al.[6], Olroyd & Sidor[7], Marchetti et al[33,36], Romano et al.[35] and De Jaime-Soguero et al.[37].

less phylogenetic and morphological diversification. This discovery opens the door for findings that may fill in the early therapsid fossil gap in the lower Permian (Fig. 4), not in high latitude sites as traditionally thought, but in the so far poorly explored lower–middle

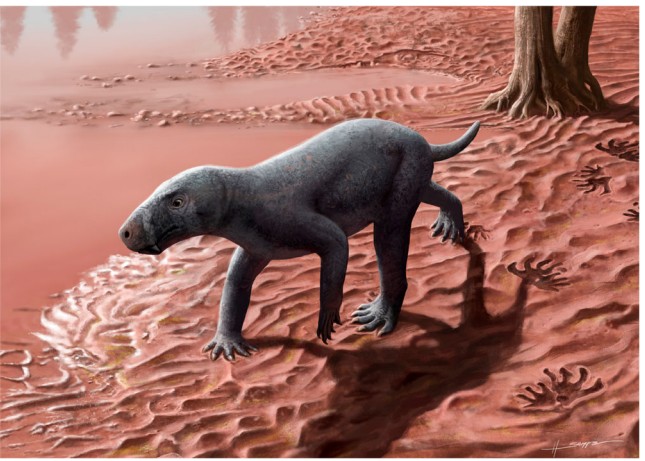

**Fig. 6 | Life reconstruction of the gorgonopsian from Mallorca in a floodplain setting.** Created by Henry Sutherland Sharpe. © 2023 Henry Sutherland Sharpe. Used under license.

Permian areas of palaeoequatorial Pangaea. Those locations hold the potential to elucidate the early evolution of therapsids and the origins of mammalian features.

## Methods

### Permits
Collection and preparation were carried out with the permission of the *Comissió Insular de Patrimoni Històric* of the *Consell Insular de Mallorca* (excavations with file numbers 306/2019 and 75/2021, preparation with file numbers 374/2020 and 295/2021).

### Data collection and curation
The specimen DA21/17-01-01 was collected from the Torrent de na Nadala site (Port des Canonge Formation, Mallorca, Balearic Islands, western Mediterranean)[9,16] (Figs. 1, 2). The tetrapod bones are very brittle and were collected in the matrix and subsequently prepared using air scribes. The bones glow under UV light (wavelength of 365 nm), which made it possible to distinguish them from the surrounding matrix, which is usually of almost the same colour. All the outcrops of this formation are located at the base of coastal cliffs in the northwestern margin of the Serra de Tramuntana, the main mountain range of the island, and have been interpreted as meandering river systems and associated floodplains deposited under a semi-arid, seasonal climate[9]. The horizon where the gorgonopsian fossil was collected (Figs. 1b, 2) consists of silty, very fine-grained, red sandstones with carbonate nodules and rhizocretions (indicating the formation of carbonate palaeosols), and fragments of coalified plant remains. Bones were scattered with no preferential orientation nor selection of elements (Fig. 2b), indicating a relatively prolonged exposure and a late stage of decay without (major) transportation of the remains[60]. In outcrop view, the fossil site corresponds to the infill of a depressed part of the floodplain, wedging laterally[16]. Considering this morphology, the very fine grain size, and the abundance of tetrapod bones (five semi-complete, partially disarticulated adult and juvenile skeletons found so far, four of them corresponding to the moradisaurine captorhinid *Tramuntanasaurus tiai*[16] and one to the gorgonopsian studied here), this deposit probably corresponded to a shrinking waterhole, similar to those grouped under the taphonomical class B1 of Smith et al.[61]. In the Karoo Basin of South Africa, waterhole deposits (corresponding to abandoned chute channels or floodplain depressions) contain abundant and well-preserved tetrapod skeletons that probably accumulated because of increased predation in such environments[62] and transportation of mummified carcasses by sheet flows[61]. In the case of Torrent de na Nadala 2,

### Legend for Fig. 5
- Tropical everwet
- Tropical summerwet
- Desert
- Mid-latitude desert
- Winterwet
- Warm temperate
- Cool temperate
- Cold temperate
- Tundra
- Mountains
- Epicontinental waters
- Oceans
- 'Pelycosaur'-grade synapsids
- Non-mammalian therapsids
- Gorgonopsian from Mallorca

another important cause of death might have been dehydration when the waterhole dried out, based on the joint presence of juvenile and adult carcasses with no apparent signs of prolonged transport or predation.

## Morphological phylogenetic dataset

To test the evolutionary hypotheses of this study, we used the only available morphological phylogenetic dataset sampling the major early therapsid clades at the species level, which was last updated by Brink et al.[45]. We modified scorings for several taxa because of differences in character interpretation from previous work. Further, we added eight species to the dataset based on personal observations of the relevant specimens and data from the literature, to provide a larger, species-level sample of the earliest known species of the major therapsid clades. The new taxa are: early sphenacodonts *Palaeohatteria longicaudata, Pantelosaurus saxonicus*, and *Cutleria wilmarthi*, the dinocephalians *Archaeosyodon praeventor* and *Tapinocaninus pamelae*, the anomodont *Ulemica efremovi*, the gorgonopsians *Viatkogorgon ivakhnenkoi, Suchogorgon golubevi, Eriphostoma microdon*, and *Arctops willistoni*, and the eutheriodont *Lycosuchus vanderrieti*. The final dataset includes a total of 39 taxa and 78 characters; the full list of changes to the dataset can be found in Supplementary Note 3. Specimen DA21/17-01-01 was not included in the analysed version of the data matrix, as it could only be coded for one character (Character 50: Dentary height in canine versus anterior postcanine regions: state 1, shows pronounced difference) and its inclusion resulted in collapse of the tree into an unresolved polytomy.

## Character evolution model

We used the Mk(v) model of character evolution[63] for morphological characters, implementing assortment bias correction for the inclusion of variable characters only. We allowed for among character rate variation by using a gamma distribution with four rate categories and with shape (alpha) sampled from an exponential distribution with mean = 1.0.

## Clock models

We assessed the fit of various clock models to our data, including a new suite of clock models available in the developer's version of Mr. Bayes (future Mr. Bayes 3.2.8 release)[64] compiled from source code available at https://github.com/NBISweden/MrBayes. These models include the continuous autocorrelated clock model (TK02)[65] and three uncorrelated clock models: the white noise model (former IGR model up to version 3.2.7a of Mr. Bayes)[66], and the new independent gamma rate (IGR) and independent lognormal clock (ILN) models[44]. These models represent radically different interpretations of how and where in the tree evolutionary rates are allowed to change across lineages, which can substantially impact divergence time estimates using morphological and/or molecular data[44,67].

We tested model fit using the stepping-stone sampling strategy to assess the marginal model likelihoods[68] and calculated Bayes Factors (BF) − 30 steps (+5 as burn in) for 250 million generations. Using the significance thresholds of Kass & Raftery[69], we found a strong support (BF > 10) for the IGR and ILN models relative to the autocorrelated (TK02) and the WN model (Supplementary Table 1), with a minor advantage for them IGR over the ILN model.

The starting value for the prior on the clock rate was given an informative prior as per previous non-clock analysis − the median value for tree height in substitutions from posterior trees divided by the age of the tree based on the median of the distribution for the root prior (5.89/307 = 0.02). The mean of the lognormal distribution was given the value based on the non-clock tree estimate in natural log scale: ln(0.02) = − 3.9535. Finally, we chose a broad standard deviation around the mean (σ = 1.0).

## Tree model and age calibrations

The age of the root was given a hard minimum age of 303.7 Ma based on minimum possible age for outgroup *Haptodus*, and a conservative soft maximum age based on the maximum estimated divergence for all Sphenacodontia at ca. 310 Ma[4] (Supplementary Data S1). We used an offset exponential distribution for the prior on root age, which gives a higher sampling probability for values closer to the minimum age and a relatively low (but nonzero) probability of age values higher than 310 Ma.

A first, analysis was conducted with tip dating only−in which all fossil calibrations (apart from the root node) were based on tip calibrations. Additionally, we avoided the strong biases that can be introduced by point age calibrations on the age of the fossils[70] by using a uniform prior distribution on the age range of the stratigraphic occurrence of the fossils (Supplementary Data S1). We implemented the skyline FBD (SFBD) process for the tree model with a flat prior (uniform distribution: samples $\in U[0,1]$) on the parameters of relative extinction (= turnover) and probability of sampling fossils, whereas the net diversification rate was sampled from an exponential distribution with mean = 1.0. These parameters were allowed to change across the Guadalupian−Lopingian boundary at 259.51 Ma, which is associated with a mass extinction, and thus provides reasonable expectation for a major shift in diversification dynamics.

In a second set of analyses, we repeated the same protocol described above, but placed a node calibration on the age of Gorgonopsia informed by the new findings reported here. This calibration implies a minimum age of 265 Ma (middle Wordian) and a maximum age of 290 Ma (Artinskian), which is the potential stratigraphic range of the beds where it was found. This node age was sampled from a uniform distribution.

## MCMC and Diagnostic parameters

All analyses were conducted using the developer's version of Mr. Bayes (future Mr. Bayes 3.2.8 release)[64] compiled from source code available at https://github.com/NBISweden/MrBayes using the Della cluster at Princeton University. Convergence of independent runs was assessed using: average standard deviation of split frequencies (ASDSF ~ 0.01), potential scale reduction factors [PSRF ≈ 1 for all parameters], and effective sample size (ESS) for each parameter greater than 200, and analysed using Tracer v. 1.7.1[71].

## Reporting summary

Further information on research design is available in the Nature Portfolio Reporting Summary linked to this article.

# Data availability

All data generated and analysed are freely available online as Supplementary Data at: https://doi.org/10.7910/DVN/9RNJAE. It contains the following: SD 1: List of changes to previous version of synapsid dataset (.doc). SD 2: Data table with list of sampled taxa including their age ranges used for time calibration in Bayesian clock analyses. SD 3: All input and output files necessary to replicate all phylogenetic analysis conducted. Input files include Mr. Bayes command block with all Bayesian inference commands with explanatory comments. The original fossil material has been deposited at the *Museu de Mallorca* (Palma, Spain), and, together with casts of all the elements, is now housed at the *MUCBO | Museu Balear de Ciències Naturals* (Sóller, Spain). The curator is Dr. Rafel Matamales-Andreu, first author of the present paper. Casts have also been deposited at the *Institut Català de Paleontologia Miquel Crusafont* (Sabadell, Spain). The collection manager is Dr. Josep Robles: josep.robles@icp.cat.

# Code availability

All code is freely available online as Supplementary Data at: https://doi.org/10.7910/DVN/9RNJAE.

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

## Acknowledgements

The authors thank Sebastià Matamalas and Chabier De Jaime for aid in fieldwork; Marina Rull and Xènia Aymerich (ICP, Cerdanyola del Vallès) for preparation of the specimen; Enric Vicens (UAB, Cerdanyola del Vallès) for granting us access to the photographic equipment of the Palaeontology Unit of the UAB; the Comissió de Patrimoni Històric del Consell Insular de Mallorca for granting us the excavation permits and funding for preparation. R.M.A. was supported by a predoctoral grant FPU17/01922 (Ministerio de Ciencia, Innovación y Universidades) and the grant Synthesys+ DE-TAF-23 (European Commission). The project was further supported by the Generalitat de Catalunya (CERCA Programme and consolidated research groups 2021 SGR 01184 to J.F. and 2021 SGR 01192 to À.G.). This work is part of the Ramon y Cajal grant to J.F. [RYC2021-032857-I] financed by MCIN/AEI/10.13039/501100011033 and the European Union 'NextGenerationEU'/PRTR. We acknowledge support from the project 'Mallorca abans dels dinosaures: estudi dels ecosistemes continentals del Permià i Triàsic amb especial èmfasi en les restes de vertebrats' (ref[15],619.), based at the Institut Català de Paleontologia Miquel Crusafont and funded by the Departament de Cultura, Patrimoni i Política Lingüística (Consell Insular de Mallorca).

## Author contributions

Conceptualization R.M.A., C.F.K., K.D.A.; Data Curation T.R.S., J.F.; Formal Analysis C.F.K., K.D.A., T.R.S.; Funding Acquisition R.M.A., À.G., J.F.; Investigation R.M.A., C.F.K., K.D.A., T.R.S.; Methodology R.M.A., C.F.K., K.D.A., T.R.S., À.G., J.F.; Project Administration J.F.; Resources À.G., J.F.; Visualisation R.M.A., T.R.S., E.M.; Writing — Original Draft Preparation R.M.A., C.F.K., T.R.S.; Writing — Review & Editing R.M.A., C.F.K., K.D.A., T.R.S., E.M., À.G., J.F.

## Competing interests

The authors declare no competing interests.
