## [Peer Review file · Nature Communications]

The oldest known gorgonopsian suggests an equatorial origin of therapsids

Corresponding Author: Dr Rafel Matamales-Andreu

Version 0:

Reviewer comments:

Reviewer #1

(Remarks to the Author)

This is an unexpected important discovery. It is worth to be published in NC but needs to be revised. Here are some of my concerns:

I agree that specimen DA21/17-01-01 is a gorgon. Its age is a big challenge here. The current evidence, especially the magnetostratigraphic one strongly supports an age older than Wordian. It is sad there is no ash-bed to produce absolute age. It could be the oldest known gorgon, but not sure the oldest therapsids. However, you need discussion on Russian Kamagorgon and Parabradyosaurus. They were referred to Gorgonopsia by some author and the age could be Rodian.

Sennikov A G, Golubev V K, 2017. Sequence of Permian Tetrapod Faunas of Eastern Europe and the Permian–Triassic Ecological Crisis. *Paleontological Journal*, 51: 600-611. 10.1134/S0031030117060077

For tip-dated calibrations on the origin of therapsids, the author should discuss the Russian tetrapod fauna such as Golyusherma and Menza, the former includes definite therapsids and the age range was proposed as early as Rodian. The age model needs to account for this.

L271 until now, tropical therapsids were rare and only found in the late Permian (Lopingian), represented by an extremely poor fossil record consisting of fragmentary gorgonopsian remains from central Africa⁴⁵ and derived dicynodonts from Laos

Wrong, during late Permian, most of the North China lay within the tropics, see

Huang B, Yan Y, Piper J D A et al. , 2018. Paleomagnetic constraints on the paleogeography of the East Asian blocks during Late Paleozoic and Early Mesozoic times. *Earth-Science Reviews*, 186: 8-36.
<https://doi.org/10.1016/j.earscirev.2018.02.004>

Biogeographic origin of therapsids

I disagree on the therapsids originating in tropical rather than temperate regions. If the age is shown to be older than Russian and Chinese faunas, it could be, or not so.

Footprints from early Permian of North China, one from BJ (in reviewing), one from Shanxi, show that North China Block already attached to Pangea ~300 ma and the tetrapods can freely migrate through current Russia. So, the possibility that the gorgon here migrated from Russia cannot be excluded right now.

The original definition of character 27 could be improved to include some information.

I disagree on some codings, e.g., Ch4, 8 in Raranimus. The latter does not matter, but the codings of ch4 are not strict according to the description, but based on comparison with other species. Raranimus is similar to Biarmosuchus in this case.

Reviewer #2

(Remarks to the Author)

I think this is a strong study that presents an important new specimen with significant ramifications for our understanding of

synapsid evolution, while also demonstrating the need for the expansion of fossil collecting efforts to under-explored areas. The manuscript is well-written, and the figures are well composed. I think the presentation of the new specimen is perhaps a little cautious, but the analytical elements of the study are quite elegant and test long-standing hypotheses on the timing and biogeography of therapsid origins. The conclusions of the authors is supported by their present results. Therefore, I am happy to support publication of this exciting manuscript at Nature Communications following minor revisions.

General Comments:

- I'm a little curious as to why a new taxon was not erected for this specimen when it is the only gorgonopsian present at this site/general location and time interval? It also seems that you have reasonable support that it's a mature individual and entirely new to science, so it seems that there's enough justification for erecting a new species here. Sure, the incompleteness of the specimen precludes its placement in the gorgonopsian phylogeny, but new taxa have been erected with less material (perhaps overzealously so); the number of Mesozoic reptiles named from limited and highly fragmentary remains springs to mind. Perhaps a little more clarification in text (picking up from line 168)?

- When referring to a particular study, I would replace only the date with the numerical in-text citation (e.g. "last updated by Smiley et al.45" "instead of last update by 45"). See line 532 (Morphological phylogenetic dataset) for this example, but this also applies to the use of "e.g." throughout the manuscript to refer to existing papers. Overall, I would generally try to reduce the use of "e.g." in the manuscript.

- Though the current findings are not unexpected and the authors should be commended for their sampling of early therapsids, I think it may be worthwhile to try rerunning the morphological clock analyses with a matrix incorporating additional sphenacodontian taxa such as Pantelosaurus, Palaeohatteria, and Cutleria to fill in the present phylogenetic gap between Haptodus and derived sphenacodontids such as Dimetrodon. This would also allow for more direct comparison with the results of Huttenlocker et al., 2021, who raised the idea of a latest Carboniferous basal synapsid-therapsid split.

- I am unfamiliar with the Bayesian skyline approach so think the present work would benefit by undergoing additional review from someone with greater expertise & experience in phylogenetic inference methods, particularly using this method.

- There doesn't yet appear to be any command scripts associated with this manuscript, but I trust the Mr Bayes commands will be supplied in a later revision and be appropriately annotated & organised.

Specific Comments:

- Lines 110-113: Some therocephalians such as Moschorhinus also have quite steep symphyses as well, so perhaps modify line 110 to "and some biarmosuchians and therocephalians...", then line 112 to "In most therocephalians and some biarmosuchians such as Herpetoskylax..."

- Lines 246 & 281-282: You mention the idea of Olson's extinction being an ecological opportunity for therapsids, and suggest a potential niche for the new specimen. In either of these instances it may be worthwhile citing Singh et al. 2024. Predatory synapsid ecomorphology signals growing dynamism of late Palaeozoic terrestrial ecosystems. Communications Biology, which discussed the ecological evolution of therapsids and the idea of early Permian competition between sphenacodontids and early therapsids, while also demonstrating an expansion of therapsid disparity and feeding functionality in the aftermath of the extinction event. However, I recognise that this is somewhat self-serving so please feel absolutely free to ignore this suggestion.

- Line 550: Typo? suit instead of suite.

- Figure 1. I think this is very strong figure, but I would suggest combining panels a & b, and reducing the size of the globe so you can give more space to expand the more important c & d panels or include additional details from the extended figure 1. I don't think panels a & b need as much space as currently given.

- Figure 2. I would perhaps reduce the main skeletal reconstruction in the centre to give more space so that you can enlarge and show the detail of the fossil elements, particularly n-w. In any case, I think it would be good to slightly increase the text size in the figure as it's a little too small at present

- Figure 4. Is the colour-scheme colour-blind friendly? Also, can you increase the opacity of the blue shading for the pelycosaur as a little strong at present and difficult to see climatic zones underneath. Maybe an outline instead of additional shading?

Reviewer #3

(Remarks to the Author)

Below is my review of the paper entitled "The oldest known gorgonopsian suggests an equatorial origin of therapsids" by Matamales-Andreu and colleagues.

This paper is outstanding—succinct, well-written, and richly illustrated—offering significant insights into the origins of therapsids. The authors have presented their findings clearly and accessibly, combining a novel and unexpected discovery with robust analytical methods to test key hypotheses about synapsid macroevolution. Their meticulous work and thorough documentation of materials and methods are commendable. It is gratifying to see such high-calibre vertebrate palaeontology

research published in a Nature journal, especially one that effectively integrates new fossil and stratigraphic data with analytical work.

I particularly appreciate their approach to incorporating new occurrences and derived analytical findings to test Kemp's hypothesis on therapsid origins. Given the authors have efficiently kept their text well within the word limit set by Nature Communications, I suggest they expand slightly on this crucial macroevolutionary topic. A brief paragraph speculating on the paleobiological phenomena—such as specific climatic adaptations that may have played out as advantageous traits during Olson's extinction, or the impact on the evolution of their thermophysiology—that might have driven therapsid radiation at that time — could be beneficial. This addition could be similar in style to the last four paragraphs of a recent paper of mine on dinosaurs (<https://doi.org/10.1016/j.cub.2024.04.051>), but should probably be kept as just a small paragraph to add where Kemp's hypothesis is discussed. I envision this addition may warrant locking in additional citations for this excellent work in the future.

I have directly annotated the word manuscript file to provide the authors with an easily accessible document containing minor edits and suggestions, either marked as track changes or comments.

Aside from these minor suggestions, I find the paper to be of exceptional quality and believe it will make a lasting contribution to vertebrate palaeontology. I congratulate the authors on their excellent work and look forward to seeing this paper published in Nature Communications.

Best regards,

Alfio Alessandro Chiarenza
University College London

Version 1:

Reviewer comments:

Reviewer #1

(Remarks to the Author)

The authors reply my concerns, although there still are a few different opinions, but the paper is okay to publish. I hope it to be publish soon.

Reviewer #2

(Remarks to the Author)

I think the authors have done well in their response to the reviewer comments as this revised manuscript is tighter and clearer, and overall, better conveys the importance of this specimen and wider relevance to synapsid evolution. I believe the authors have given a thorough response to the reviewer comments, making important additions and clarifications to the study and manuscript. I am glad to see that they have followed most of my recommendations, and it's clear to see that the changes have been made with good effort. They have provided detailed replies to all my comments and I am happy with their responses and the revised manuscript. Therefore, I am pleased to endorse its publication once the following minor revisions are made:

Main Comment:

Introduction - I think it's worthwhile trying to mention Olson's extinction in the abstract, if possible, but certainly in the introduction. Raising how this specimen relates to the potential ecological opportunity of Olson's extinction in the intro further highlights its importance in providing greater eco-evolutionary context on the origin and early radiation of therapsids. Moreover, it also better ties this paper to a broader core topic in evolutionary biology, expanding its appeal across life science readers. I think brief mention of this would suffice.

Minor Changes:

Abstract - Rephrase: For clarity, I would amend "from Mallorca, western Mediterranean" to "from the island of Mallorca in the western Mediterranean", if the word limit permits.

Line 150 - Typo: "with increasing in body size" change to either "with increases in body size" or "increasing body size".

Line 258 - Rephrase: I would change "morphotypes in terrestrial prey animals" to "morphotypes in terrestrial animals" as this encompasses the changes in both predators and prey during this time. You elaborate on the prey later in the discussion so I think it works better here to be more general as both groups show morphological changes.

Line 271-2 – Rephrase: I would change the line, "regions such as the continental northwestern domain" to "regions from the continental northwestern Tethyan domain", to be clear about which areas you're referring to.

Line 333 – Rephrase: I would delete "now" from, "with therapsids now constituting a key new component" as I think this change makes it clear that you're still referring to the early Permian and avoids any additional cause for confusion over timing.

I look forward to seeing this work published and congratulate the authors on a job well done.

Reviewer #3

(Remarks to the Author)

I think that the revised MS is in great shape and I am particularly happy with their implemented paragraph re: the possible connection of therapsids's palaeobiology, their invasion of the tropic and patterns of survival following Olson's extinction. At a general read I think the MS is in great shape and pretty much ready to be published as it is in Nat Comms.

Kinds regards,

Alfio Alessandro Chiarenza
University College London

Responses to Reviewer #1 (anonymous):

This is an unexpected important discovery. It is worth to be published in NC but needs to be revised. Here are some of my concerns:

I agree that specimen DA21/17-01-01 is a gorgon. Its age is a big challenge here. The current evidence, especially the magnetostratigraphic one strongly supports an age older than Wordian. It is sad there is no ash-bed to produce absolute age. It could be the oldest known gorgon, but not sure the oldest therapsids. However, you need discussion on Russian Kamagorgon and Parabradyosaurus. They were referred to Gorgonopsia by some author and the age could be Rodian.

> The only works to consider *Kamagorgon* and *Parabradyosaurus* as gorgonopsians are the monographs of the late Mikhail Ivakhnenko (2003, 2008). Tatarinov (1999) originally described *Kamagorgon* as an ‘eotitanosuchid’ (= biarmosuchian under modern classification). *Parabradyosaurus* was originally described as a pareiasaur, but it has been recognised as a herbivorous dinocephalian since at least the work of Olson (1962). Ivakhnenko’s ideas of synapsid relationships were unorthodox (among other things, he considered monotremes to be the living descendants of dicynodonts). His expansive conception of Gorgonopsia included not only *Kamagorgon* and *Parabradyosaurus*, but also almost the entirety of Biarmosuchia, as well as *Biarmosuchoidea* (a probable therocephalian—see Kammerer, 2023; Suchkova *et al.*, 2023), and *Estemmenosuchus* (otherwise universally recognised as a dinocephalian). Ivakhnenko’s hypotheses were based on only a few, plesiomorphic characters, and he did not carry out any analytical phylogenetic testing. For example, his entire diagnosis for gorgonopsians was: “Temporal fenestra developed mainly posterosuperiorly; therefore, upper region of occipital plate of squamosal curved posteriorly. Temporal fenestra almost lacking anterodorsal expansion; dorsoexternally, anterior part of temporal fenestra usually covered by postorbital” (Ivakhnenko, 2003: p. S392). However, this combination of characters describes any primitive therapsid temporal fenestra that lacks a medial expansion creating a sagittal crest (evolved independently in anomodonts, eutheriodonts, and anteosaurian dinocephalians). No recent therapsid literature (see reviews by Rubidge & Sidor, 2001; Angielczyk & Kammerer, 2018) considers these taxa to be gorgonopsians and we believe their inclusion in this manuscript to be both unnecessary and potentially misleading, requiring a lengthy digression into the chequered taxonomic history of these obscure fossils. As a final note, although these species have received limited coverage in the literature, their holotypes have been examined personally by one of us (C. Kammerer), finding that they exhibit no gorgonopsian characters.

References:

- Angielczyk, K.D., and Kammerer, C.F. 2018. Non-mammalian synapsids: the deep roots of the mammalian family tree. Chapter 5 (pp. 117-198) in Asher, R., and F. Zachos (eds.) *Handbook of Zoology. Mammalia*. Berlin: DeGruyter.
- Ivakhnenko, M.F. 2003. Eotherapsids from the East European Placket (Late Permian). *Paleontological Journal* 37: S339-S465.
- Ivakhnenko, M.F. 2008. Cranial morphology and evolution of Permian Dinomorpha (Eotherapsida) of Eastern Europe. *Paleontological Journal* 42: 859-995.
- Kammerer, C.F. 2023. Revision of the Scylacosauridae (Therapsida: Therocephalia). *Palaeontologia africana* 56: 51-87.

- Olson, E.C. 1962. Late Permian terrestrial vertebrates, U.S.A. and U.S.S.R. *Transactions of the American Philosophical Society* 52: 1-224.
- Rubidge, B.S., and Sidor, C.A. 2001. Evolutionary patterns among Permo-Triassic therapsids. *Annual Review of Ecology and Systematics* 32: 449-480.
- Suchkova, Y.A., Golubev, V.K., and Shumov, I.S. 2023. New primitive therocephalians from the Permian of Eastern Europe. *Paleontological Journal* 56: 1419-1427.
- Tatarinov, L.P. 1999. A new eotitanosuchid (Reptilia, Therapsida) from the Kazanian Stage (Upper Permian) of Udmurtia. *Paleontological Journal* 33: 660-666.

Sennikov A G, Golubev V K, 2017. Sequence of Permian Tetrapod Faunas of Eastern Europe and the Permian–Triassic Ecological Crisis. *Paleontological Journal*, 51: 600-611. 10.1134/S0031030117060077

For tip-dated calibrations on the origin of therapsids, the author should discuss the Russian tetrapod fauna such as Golyusherma and Menza, the former includes definite therapsids and the age range was proposed as early as Rodian. The age model need account for this.

> The Golyusherma Fauna is considered the stratigraphically oldest Russian therapsid assemblage; as the reviewer notes, it might be as old as Roadian based on biostratigraphy (but it could be also younger, because no absolute date exists and its vertebrate fauna is very similar to that of probable Wordian assemblages). Its known synapsid components consist entirely of dinocephalians, which are represented solely by jaw fragments. Those that have been included in phylogenetic analyses (e.g., *Microsyodon orlovi* in the analysis of Kammerer, 2011) have been found to be no older (vs. *Sinophoneus* from the Chinese Dashankou assemblage) or more basal (vs. *Archaeosyodon* from the Russian Ocher assemblage) than taxa known from much more complete remains that we have included in our current analysis. Also, those Golyusherma taxa that are too fragmentary to have been included in any previous analyses show close similarity to taxa that we included here (e.g., the jaw fragment of *Parabradysaurus* is extremely similar to that of *Estemmenosuchus*, to the point that Olson (1962) suggested that they could be synonymous). So, adding Golyusherma taxa to the analysis would not alter our age model, because we already include equally if not more ancient/basal exemplars for every major clade. Given their highly fragmentary nature, their inclusion as tips would increase uncertainty in phylogenetic estimation. Our goal is not to include every single synapsid species in our analysis and tentatively place fragmentary taxa, but to include highly informative taxa and use their completeness and stratigraphic and phylogenetic importance to provide a robust time-tree.

Mezen is a very unusual assemblage numerically dominated by ‘parareptiles’ and ‘pelycosaur’, which is usually considered Wordian in age (e.g., Benton, 2012; although again no precise radioisotopic dates exist). This would not make the minimum age of therapsids older than the minimum age we already provided in our model. Older (Roadian) representatives are known for the three earliest-diverging therapsid clades in our phylogeny (*Raranimus*, Dinocephalia and Anomodontia) and a comparable age is hypothesised for the earliest-diverging biarmosuchian (*Biarmosuchus*), again having no impact on our calibration points.

Phylogenetic placement of synapsids occurring in the Mezen assemblage is hampered by their ontogenetic status. The most complete therapsid fossils from this assemblage are the ‘nikkasaurids’ *Nikkasaurus* and *Reiszia*, represented by almost complete skulls and in some cases associated postcrania, which are almost certainly very young juveniles (based on their enormous

orbits, undifferentiated dentition, unfused calvaria and vertebrae, and unfinished ends of their long bones). Based on gestalt (Kammerer, pers. obs. of all known nikkosaurid specimens) these are probably the juveniles of dinocephalians, although including them in phylogenetic analyses is complicated by the tendency of specimens at very early ontogenetic stages to move away from adults ('ontogeny discombobulating phylogeny'; Wiens *et al.*, 2005) and the absence of most of the cranial ornamentation features used in dinocephalian phylogeny in these poorly-ossified specimens.

In conclusion, the Mezen and Golyusherma therapsid faunas are made up of specimens that are historically systematically ambiguous (with this being driven by incompleteness or ontogenetic stage) and which have no impact on our age calibrations (even assuming their oldest possible age). A discussion about them in the context of the present manuscript would perhaps be of interest to specialists on the Russian therapsid fauna, but it would be a digression from the central point of this manuscript.

References:

- Benton, M.J. 2012. No gap in the Middle Permian record of terrestrial vertebrates. *Geology* 40: 339-342.
- Kammerer, C.F. 2011. Systematics of the Anteosauria (Therapsida: Dinocephalia). *Journal of Systematic Palaeontology* 9: 261-304.
- Wiens, J.J., Bonett, R.M., and Chippindale, P.T. 2005. Ontogeny discombobulates phylogeny: Paedomorphosis and higher-level salamander relationships. *Systematic Biology* 54: 91-110.

L271 until now, tropical therapsids were rare and only found in the late Permian (Lopingian), represented by an extremely poor fossil record consisting of fragmentary gorgonopsian remains from central Africa⁴⁵ and derived dicynodonts from Laos

Wrong, during late Permian, most of the North China lay within tropic, see

Huang B, Yan Y, Piper J D A et al. , 2018. Paleomagnetic constraints on the paleogeography of the East Asian blocks during Late Paleozoic and Early Mesozoic times. *Earth-Science Reviews*, 186: 8-36. <https://doi.org/10.1016/j.earscirev.2018.02.004>

> We have now added some lines accounting for this in the Discussion. Briefly, although the North China Block is currently best reconstructed as being tropical in the Permian, what is now northern China was in the Permian made up of a series of separate landmasses that were not attached to this block until the Triassic or later. Most late Permian synapsid fossils are from Xinjiang, which was part of a 'Jungar Block' or 'Junggaria', but more importantly, the Roadian Dashankou fauna is from a region of Gansu (near Yumen City) that was part of a separate terrane (called Alashan or Alxa) that recent papers indicate existed to the north of the North China Block and may have been subtropical. Although we do think it is possible for therapsids to be present on the North China Block at this time (and indeed it would support our arguments for their initial diversification occurring in palaeoequatorial latitudes), we nevertheless believe we are accurately representing our new discovery as the earliest unequivocal palaeoequatorial therapsid, given conflicting reconstructions of China's assembly in the late Palaeozoic and the precise location (in other blocks) of most Chinese therapsids.

We have modified the text adding the following paragraph (lines 289-295):

‘Richer therapsid assemblages are known from northern China, but the complex geological history and conflicting palaeogeographic reconstructions for this region complicates confident palaeolatitudinal assignments. Although current evidence indicates that the North China Block was within the tropics during the Permian⁴⁹, the portion of Gansu yielding middle Permian therapsids (the possibly Roadian Dashankou assemblage) was part of a distinct land mass (the Alashan Terrane) that did not amalgamate with the North China Block until the Mesozoic and was situated north of it (though at least in the subtropics)⁵⁰.’

Biogeographic origin of therapsids

I disagree on the therapsids originated in tropical rather than temperate regions. If the age is showed to be older than Russian and Chinese faunas, it could be, or not so.

Footprints from early Permian of North China, one from BJ (in reviewing), one from Shanxi, show that North China Block already attached to Pangea ~300 ma and the tetrapods can freely migrate through current Russia. So, the possibility that the gorgon here migrated from Russia cannot be excluded right now.

> Indeed, the paper by Chen & Liu (2024) suggests that the North China block was already connected to Pangaea during the *Dromopus* Biochron (late Carboniferous–early Permian). We understand the reviewer’s train of thought that if tetrapods could freely migrate between Russia and China, then they could also freely migrate from Russia to Mallorca, or from China to Mallorca via Russia. However, we have to bear in mind three facts:

(1): A Roadian–early Wordian age for the Mallorcan gorgonopsian is the most conservative age estimate. As pointed out in the manuscript, all the age data we have (palaeomagnetism, palaeopalinology, ichnostratigraphy) point to an Artinskian–Kungurian age, which is even older than our already conservative minimal age calibration. Please see the first paragraph of the section ‘Age of the oldest known therapsids’.

(2): Bearing this in mind, there is a very high chance that the Mallorcan gorgonopsian is the oldest therapsid known, older than the described Roadian faunas of Russia and China.

(3): Among the tetrapod tracks illustrated by Chen & Liu (2024), none of them was putatively produced by a therapsid.

Therefore, based on all information currently available, the (very likely) oldest therapsid being found on Mallorca strongly suggests that therapsids originated in palaeoequatorial latitudes. This would also align with older hypotheses on the origin of therapsids (Kemp and others), and would fill the biogeographical gap between the northern and southern middle and late Permian therapsid faunas. We have modified the text to clarify this point. Specifically, the following changes have been made:

‘possibly’ → ‘**very likely**’ (line 59)

‘must be older than middle Wordian’ → ‘**is certainly older than middle Wordian**’ (lines 197-198)

‘whereas a Kungurian or Artinskian age would make it the oldest definitive therapsid discovered to date.’ → ‘whereas a Kungurian or Artinskian age, which best accords with the magnetostratigraphic and ichnological data, would make it the oldest definitive therapsid discovered to date.’ (lines 206-208)

The original definition of character 27 could be improved to include some information.

> We considered this character problematic in the context of this analysis because all coded taxa show identical morphologies for it. If we changed it, it would be to delete it entirely, but as even non-parsimony-informative characters have importance in the Bayesian models we are using, we decided to leave it as-is, with all taxa coded the same.

I disagree on some codings, e.g., Ch4, 8 in *Raranimus*. The latter does not matter, but the codings of ch4 are not strict according to the description, but based on comparison with other species. *Raranimus* is similar to *Biarmosuchus* in this case.

> Character 4 is not relative between species; it is determinable within any skull examined in isolation (including *Raranimus*) and defined by whether the dorsal process of the premaxilla extends posterior to the upper canine or not. Similarly, character 8 is a simple yes/no: does the maxilla contact the prefrontal or not. We do not believe there are alternative ways to code these characters.

Responses to Reviewer #2 (anonymous):

I think this is a strong study that presents an important new specimen with significant ramifications for our understanding of synapsid evolution, while also demonstrating the need for the expansion of fossil collecting efforts to under-explored areas.

The manuscript is well-written, and the figures are well composed. I think the presentation of the new specimen is perhaps a little cautious, but the analytical elements of the study are quite elegant and test long-standing hypotheses on the timing and biogeography of therapsid origins. The conclusions of the authors is supported by their present results. Therefore, I am happy to support publication of this exciting manuscript at Nature Communications following minor revisions.

General Comments:

- I'm a little curious as to why a new taxon was not erected for this specimen when it is the only gorgonopsian present at this site/general location and time interval? It also seems that you have reasonable support that it's a mature individual and entirely new to science, so it seems that there's enough justification for erecting a new species here. Sure, the incompleteness of the specimen precludes its placement in the gorgonopsian phylogeny, but new taxa have been erected with less material (perhaps overzealously so); the number of Mesozoic reptiles named from limited and highly fragmentary remains springs to mind. Perhaps a little more clarification in text (picking up from line 168)?

> We prefer to err on the side of caution when erecting new taxa. We consider that there must be at least a single unique morphological character (*i.e.*, autapomorphy) that allows a new taxon to be distinguished from other taxa. Unfortunately, the Mallorcan gorgonopsian shows no such features—instead, it presents a number of characters that make it certain that it is a gorgonopsian, but it is not complete enough to be differentiated at lower taxonomic levels. Thus, a new taxon might be erected with the discovery of additional fossils, but a taxon named on the currently available material would represent a *nomen dubium*.

- When referring to a particular study, I would replace only the date with the numerical in-text citation (*e.g.* “last updated by Smiley *et al.*⁴⁵” “instead of last update by ⁴⁵”). See line 532 (Morphological phylogenetic dataset) for this example, but this also applies to the use of “*e.g.*” throughout the manuscript to refer to existing papers. Overall, I would generally try to reduce the use of “*e.g.*” in the manuscript.

> Agreed. The citations have been changed as the reviewer suggests. The use of ‘*e.g.*’ has been reduced throughout the manuscript when referring to citations. It has been left as it was when we refer to genera or species, because they are just examples among several therapsids that share the same trait(s). The changes are listed as follows:

‘fossil record (*e.g.*,¹¹)’ → ‘fossil record⁸’ (line 53)
‘(*e.g.*,²³: fig. 206)’ → ‘(Sigogneau²²: fig. 206)’ (line 122)
‘pterygoid (*e.g.*, lycosuchids²¹)’ → ‘pterygoids²¹’ (lines 129-130)
‘character polarity of²⁸’ → ‘character polarity of Kammerer & Masyutin²⁷’ (lines 158-159)
‘*sensu*²⁸’ → ‘*sensu* Kammerer & Masyutin²⁷’ (line 168)
‘data matrix of⁴⁴’ → ‘data matrix of Brink *et al.*⁴²’ (line 226)
‘known as Olson’s extinction (*e.g.*,⁴⁶⁻⁴⁸)’ → ‘known as Olson’s extinction at ca. 273 Ma⁴³⁻⁴⁵’ (lines 249-250)
‘(*e.g.*,⁴¹)’ → ‘(*e.g.*, Spindler³⁹)’ (lines 270-271)
‘from¹²’ → ‘from Matamales-Andreu *et al.*⁹’ (line 618)
‘Maps after Scotese⁴⁷, and climate zones after ⁴⁸ for the early and middle Permian, and ⁴⁹ for the late Permian. Synapsid distribution after^{4,5,10,32,34,36}’ → ‘Maps after Scotese⁴⁷, and climate zones after Rees *et al.*⁴⁸ for the early and middle Permian, and Smith *et al.*⁴⁹ for the late Permian. Synapsid distribution after Laurin *et al.*⁵, Bernardi *et al.*⁶, Olroyd & Sidor⁷, Marchetti *et al.*^{10,31}, and Romano *et al.*³³’ (lines 663-665)
‘taphonomical class B1 of⁶³’ → ‘taphonomical class B1 of Smith *et al.*⁵²’ (lines 687-688)
‘last updated by⁴⁵’ → ‘last updated by Brink *et al.*⁴²’ (line 707)
‘thresholds of⁶⁰’ → ‘thresholds of Kass & Raftery⁷¹’ (line 739)

- Though the current findings are not unexpected and the authors should be commended for their sampling of early therapsids, I think it may be worthwhile to try rerunning the morphological clock analyses with a matrix incorporating additional sphenacodontian taxa such as *Pantelosaurus*, *Palaeohatteria*, and *Cutleria* to fill in the present phylogenetic gap between *Haptodus* and derived sphenacodontids such as *Dimetrodon*. This would also allow for more direct comparison with the results of Huttenlocker *et al.*, 2021, who raised the idea of a latest Carboniferous basal synapsid-therapsid split.

> We added these taxa to the analysis, as requested by the reviewer, but this change did not impact our final results (Fig. 3 and Extended Data Fig. 7).

- There doesn't yet appear to be any command scripts associated with this manuscript, but I trust the Mr Bayes commands will be supplied in a later revision and be appropriately annotated & organised.

> All the necessary input data and output files (including Mr. Bayes blocks) are now freely available on a public repository that will be published with the manuscript (check details in README file).

Reviewers can access them through:

<https://dataverse.harvard.edu/privateurl.xhtml?token=1b6d0852-1b6d-43a8-8ee4-850114e368ae>

We have added the data and code availability statements text in the manuscript:

'Data availability statement. All data generated and analysed are freely available online as Supplementary Data at: <https://dataverse.harvard.edu/privateurl.xhtml?token=1b6d0852-1b6d-43a8-8ee4-850114e368ae> [permanent DOI to be provided pending acceptance].

Code availability statement. All code is freely available online as Supplementary Data at: <https://dataverse.harvard.edu/privateurl.xhtml?token=1b6d0852-1b6d-43a8-8ee4-850114e368ae> [permanent DOI to be provided pending acceptance]'. (lines 777-785)

Specific Comments:

- Lines 110-113: Some therocephalians such as *Moschorhinus* also have quite steep symphyses as well, so perhaps modify line 110 to “and some biarmosuchians and therocephalians...”, then line 112 to “In most therocephalians and some biarmosuchians such as *Herpetoskylax*...”

> Agreed. The sentences mentioned have been changed according to their suggestion:

‘A steep mandibular symphysis is also present in anteosaurs²⁰, and some biarmosuchians (e.g., *Hipposaurus*¹⁹), but in these taxa no diastema exists between the lower canine and the (usually extensive) postcanine tooth row. In therocephalians the symphysis is generally weakly sloping, with a low, convex anterior face²¹; this is also true of some biarmosuchians (e.g., *Herpetoskylax*²²).’ → ‘A steep mandibular symphysis is also present in anteosaurs¹⁸, and some biarmosuchians and therocephalians (e.g., *Hipposaurus*¹⁷, *Moschorhinus*¹⁹), but in these taxa no diastema exists between the lower canine and the (usually extensive) postcanine tooth row. In most therocephalians and some biarmosuchians (e.g., *Herpetoskylax*²⁰), the symphysis is generally weakly sloping, with a low, convex anterior face²¹.’ (lines 112-117)

- You mention the idea of Olson's extinction being an ecological opportunity for therapsids, and suggest a potential niche for the new specimen. In either of these instances it may be worthwhile citing Singh *et al.* 2024. Predatory synapsid ecomorphology signals growing dynamism of late Palaeozoic terrestrial ecosystems. *Communications Biology*, which

discussed the ecological evolution of therapsids and the idea of early Permian competition between sphenacodontids and early therapsids, while also demonstrating an expansion of therapsid disparity and feeding functionality in the aftermath of the extinction event. However, I recognise that this is somewhat self-serving so please feel absolutely free to ignore this suggestion.

> Agreed. We are very thankful to the reviewer for this idea, and we have incorporated it when we talk about Olson's extinction:

'This raises the possibility that the loss of these competitors provided ecological opportunity for emerging therapsid clades, catalysing their evolutionary radiation in the middle Permian.' → 'After this event, the emergence of robust herbivorous lineages, such as pareiasaurs, appears to have prompted a dramatic shift in jaw morphofunctionality of carnivorous synapsids (mostly therapsids) towards injuring and subduing the prey, instead of the optimisation for grasping and holding smaller prey seen in 'pelycosaurs'⁴⁶. The loss of potential competitors and the evolution of novel morphotypes in terrestrial prey animals may have provided ecological opportunity for emerging therapsid clades, catalysing their evolutionary radiation in the middle Permian.' (lines 253-260)

Comment 7: 'Line 550: Typo? suit instead of suite'

> It was indeed a typo. We have changed 'suit' to 'suite' (line 728).

Comment 8: 'Figure 1. I think this is very strong figure, but I would suggest combining panels a & b, and reducing the size of the globe so you can give more space to expand the more important c & d panels or include additional details from the extended figure 1. I don't think panels a & b need as much space as currently given'

> Agreed. We have merged panels a and b, panel c has become the new panel b, and we have removed panel d, featuring the previous panel c from Extended figure 1 instead.

Comment 9: 'Figure 2. I would perhaps reduce the main skeletal reconstruction in the centre to give more space so that you can enlarge and show the detail of the fossil elements, particularly n-w. In any case, I think it would be good to slightly increase the text size in the figure as it's a little too small at present'

> Thank you very much for this comment. Since all the elements are thoroughly illustrated in the Supplementary Material, we believe that this figure should highlight the silhouette with all the bones positioned. We think that by doing so, we can also attract a broader readership and improve the paper's impact in general. We would also like to retain the font size, as it complies with the journal's policies and the text of the rest of figures is at the same size.

Comment 10: 'Figure 4. Is the colour-scheme colour-blind friendly? Also, can you increase the opacity of the blue shading for the pelycosaurs as a little strong at present and difficult to see climatic zones underneath. Maybe an outline instead of additional shading?'

> The figure was not colour-blind friendly indeed. To fix this, we have changed the colours of the climate zones to a single colour: purple, with a lighter or darker hue depending on the climate. The shading of the distribution areas of ‘pelycosaurs’ and therapsids has been changed to two different outlines, as the reviewer suggests. Their line styles are also colour-blind friendly.

Responses to Reviewer #3 (Dr. Afio Alessandro Chiarenza):

Dear Editor,

Below is my review of the paper entitled “The oldest known gorgonopsian suggests an equatorial origin of therapsids” by Matamales-Andreu and colleagues. This paper is outstanding—succinct, well-written, and richly illustrated—offering significant insights into the origins of therapsids. The authors have presented their findings clearly and accessibly, combining a novel and unexpected discovery with robust analytical methods to test key hypotheses about synapsid macroevolution. Their meticulous work and thorough documentation of materials and methods are commendable. It is gratifying to see such high-calibre vertebrate palaeontology research published in a Nature journal, especially one that effectively integrates new fossil and stratigraphic data with analytical work.

I particularly appreciate their approach to incorporating new occurrences and derived analytical findings to test Kemp's hypothesis on therapsid origins. Given the authors have efficiently kept their text well within the word limit set by Nature Communications, I suggest they expand slightly on this crucial macroevolutionary topic. A brief paragraph speculating on the paleobiological phenomena—such as specific climatic adaptations that may have played out as advantageous traits during Olson’s extinction, or the impact on the evolution of their thermophysiology—that might have driven therapsid radiation at that time — could be beneficial. This addition could be similar in style to the last four paragraphs of a recent paper of mine on dinosaurs (<https://doi.org/10.1016/j.cub.2024.04.051>), but should probably be kept as just a small paragraph to add where Kemp’s hypothesis is discussed. I envision this addition may warrant locking in additional citations for this excellent work in the future.

> We have added a paragraph dealing with this topic, relating it to morphofunctional changes concurrently observed related to prey capture in the involved lineages as part of the broader suite of potential drivers of therapsid diversification:

‘If therapsids originated in the tropics, this has implications for metabolic evolution in the clade. Although recent evidence indicates that mammal-like endothermy in synapsids did not originate until the Triassic⁵³. The expansion of early therapsids in higher latitudes⁸ suggests that, following Olson’s extinction, the members of this clade were more eurythermal due to some rudimentary form of thermoregulation. The palaeotropical gorgonopsian presented here implies that this early thermal niche relaxation was probably the result of an exaptive process, in which early therapsids first developed their revolutionary new suite of morphological and physiological traits (more efficient locomotion, respiration, food manipulation, and elevated growth rates, among others) in the ancestral synapsid equatorial belt, which later facilitated their colonisation of cooler climates. Given that the earliest therapsids were all faunivorous⁵⁴, an additional selective force operating

on the therapsid body plan may have been the appearance of larger prey that required novel methods to subdue⁴⁶. Large moradisaurine captorhinids became abundant in the Kungurian–Roadian^{55,56}, with one of these species and large tracks correlated to this clade found in the same formation as the Mallorcan gorgonopsian^{9,57}, and geographically proximate beds of similar age in Menorca, Sardinia and southern France have yielded enormous moradisaurine captorhinids⁵⁸ and caseid ‘pelycosaurs’⁵⁹. Possibly, the traits selected to prey upon these animals also benefitted therapsids as they expanded into new environments inhabited by other large tetrapods.’ (lines 307-323)

I have directly annotated the word manuscript file to provide the authors with an easily accessible document containing minor edits and suggestions, either marked as track changes or comments.

Aside from these minor suggestions, I find the paper to be of exceptional quality and believe it will make a lasting contribution to vertebrate palaeontology. I congratulate the authors on their excellent work and look forward to seeing this paper published in Nature Communications.

Best regards,

Alfio Alessandro Chiarenza

University College London

> Agreed. Most of the reviewer’s suggestions have been applied as follows:

‘badly-flattened’ → ‘**highly-flattened**’ (line 87)

‘i4/c1/pc5’ → ‘**i4/c1/pc5 (incisor/canine/postcanine)**’ (line 94)

‘also the caniniforms of some sphenacodontids’ → **left as it was** (line 122)

‘we used for the first time an explicit statistical approach to estimate divergence times’ → ‘**we used for the first time mechanistic models of evolution to estimate divergence times**’ (lines 121-122)

‘collecting efforts in under-sampled regions such as Mallorca’ → ‘**collecting efforts in under-sampled regions such as the continental northwestern Tethyan domain**’ (lines 259-260)

Comment in extended data fig. 7 caption: ‘Maybe move the tip names more towards the right of the panel to avoid overlapping with the HPD intervals’

> Thank you for the suggestion. Unfortunately, to completely avoid the HPD bars the names would have to be too far away from the tree edges, making it harder to see where each taxon goes and giving the impression that taxa are actually younger (names become displaced towards a much younger side of the time-tree). But we have corrected that by changing the transparency values of the purple bars.

Reviewer #1 (anonymous):

The authors reply my concerns, although there still are a few different opinions, but the paper is okay to publish. I hope it to be publish soon.

Thank you.

Reviewer #2 (anonymous):

I think the authors have done well in their response to the reviewer comments as this revised manuscript is tighter and clearer, and overall, better conveys the importance of this specimen and wider relevance to synapsid evolution. I believe the authors have given a thorough response to the reviewer comments, making important additions and clarifications to the study and manuscript. I am glad to see that they have followed most of my recommendations, and it's clear to see that the changes have been made with good effort. They have provided detailed replies to all my comments and I am happy with their responses and the revised manuscript. Therefore, I am pleased to endorse its publication once the following minor revisions are made:

Main Comment:

Introduction - I think it's worthwhile trying to mention Olson's extinction in the abstract, if possible, but certainly in the introduction. Raising how this specimen relates to the potential ecological opportunity of Olson's extinction in the intro further highlights its importance in providing greater eco-evolutionary context on the origin and early radiation of therapsids. Moreover, it also better ties this paper to a broader core topic in evolutionary biology, expanding its appeal across life science readers. I think brief mention of this would suffice.

> Thank you for this suggestion. A brief remark has been added in the Abstract: "[...] evolutionary radiation of all major therapsid clades within less than 10 Myr, in the aftermath of Olson's extinction" (line 29). In the introduction, we have also added a sentence referring to this: "Olson's Extinction, which happened in the early-middle Permian transition¹¹⁻¹³, may have provided the necessary ecological opportunity for this previously inconspicuous group to diversify into varied phenotypes and distinct ecological roles." (lines 59-61).

Minor Changes:

Abstract - Rephrase: For clarity, I would amend "from Mallorca, western Mediterranean" to "from the island of Mallorca in the western Mediterranean", if the word limit permits.

> We changed it to "from the island of Mallorca, western Mediterranean" (lines 24-25), because this way we are at 149 words, and therefore we cannot add the words "in the". Regardless, we think it is easier to understand now.

Line 150 - Typo: "with increasing in body size" change to either "with increases in body size" or "increasing body size".

> Applied. Thank you.

Line 258 - Rephrase: I would change "morphotypes in terrestrial prey animals" to

“morphotypes in terrestrial animals” as this encompasses the changes in both predators and prey during this time. You elaborate on the prey later in the discussion so I think it works better here to be more general as both groups show morphological changes.

> Applied. Thank you.

Line 271-2 – Rephrase: I would change the line, “regions such as the continental northwestern domain” to “regions from the continental northwestern Tethyan domain”, to be clear about which areas you’re referring to.

> Applied. Thank you.

Line 333 – Rephrase: I would delete “now” from, “with therapsids now constituting a key new component” as I think this change makes it clear that you're still referring to the early Permian and avoids any additional cause for confusion over timing.

> Applied. Thank you.

I look forward to seeing this work published and congratulate the authors on a job well done.

Reviewer #2 (Remarks on code availability):

The supplementary info is nicely organised and comprehensive. The text files appear to indicate that all required files are present, but I admit that I have not run the scripts myself as I am unfamiliar with Mr. Bayes.

Thank you.

Reviewer #3 (Alfio Alessandro Chiarenza):

I think that the revised MS is in great shape and I am particularly happy with their implemented paragraph re: the possible connection of therapsids's palaeobiology, their invasion of the tropic and patterns of survival following Olson's extinction. At a general read I think the MS is in great shape and pretty much ready to be published as it is in Nat Comms.

**Kinds regards,
Alfio Alessandro Chiarenza
University College London**

Thank you.